# Role of Rendezvous-Procedure in the Treatment of Complications after Laparoscopic Sleeve Gastrectomy

**DOI:** 10.3390/jcm10235670

**Published:** 2021-11-30

**Authors:** Dörte Wichmann, Veit Scheble, Stefano Fusco, Ulrich Schweizer, Felix Hönes, Wilfried Klingert, Alfred Königsrainer, Rami Archid

**Affiliations:** 1Department of General and Transplant Surgery, University Hospital of Tübingen, 72076 Tübingen, Germany; doerte.wichmann@med.uni-tuebingen.de (D.W.); ulrich.schweizer@med.uni-tuebingen.de (U.S.); felix.hoenes@med.uni-tuebingen.de (F.H.); wilfried.klingert@med.uni-tuebingen.de (W.K.); alfred.koenigsrainer@med.uni-tuebingen.de (A.K.); rami.archid@med.uni-tuebingen.de (R.A.); 2Department of Internal Medicine 1, Division for Gastroenterology, Hepatology, Infectiology, Gastrointestinal Oncology and Geriatrics, University Hospital of Tübingen, 72076 Tübingen, Germany; veit.scheble@med.uni-tuebingen.de

**Keywords:** bariatric surgery, postsurgical complication management, endoscopic negative pressure therapy

## Abstract

Introduction: Laparoscopic sleeve gastrectomy is one of the most commonly performed bariatric procedures worldwide with good results, high patient acceptance, and low complication rates. The most relevant perioperative complication is the staple line leak. For the treatment of this complication, endoscopic negative pressure therapy has proven particularly effective. The correct time to start endoscopic negative pressure therapy has not been the subject of studies to date. Methods: Twelve patients were included in this retrospective data analysis over three years. Endoscopic negative pressure therapy was carried out using innovative open pore suction devices. Patients were treated with simultaneous surgery and endoscopy, so called rendezvous-procedure (Group A) or solely endoscopically, or in sequence surgically and endoscopically (Group B). Therapy data of the procedures and outcome measures, including duration of therapy, therapy success, and change of treatment strategy, were collected and analysed. Results: In each group, six patients were treated (mean age 52.96 years, 4 males, 8 females). Poor initial clinical situation, time span of endoscopic negative pressure therapy (Group A 31 days vs. Group B 18 days), and mean length of hospital stay (Group A 39.5 days vs. Group B 20.17 days) were higher in patients with rendezvous procedures. One patient in Group B died during the observation time. Discussion: Rendezvous procedures for patients with staple line leaks after sleeve gastrectomy is indicated for serious ill patients with perigastric abscesses and in need of laparoscopic lavage. The one-stage complication management with the rendezvous procedure seems not to result in an obvious advantage in the further outcome in patients with staple line leaks after laparoscopic sleeve gastrectomy.

## 1. Introduction

The number of surgeries and metabolic interventions for patients with obesity have increased worldwide [1,2]. The majority of bariatric interventions are performed surgically, especially laparoscopically [3,4]. The most common surgery is the laparoscopic sleeve gastrectomy (LSG) to minimize the volume of the stomach and thus reduce food intake [5,6]. This surgical intervention is easy to implement, as it does not contain any anastomose. Surgical complications after sleeve gastrectomy remain challenging, especially the management of staple line leaks (SLL), which occur in up to 2% of LSG patients [7]. Early diagnosis of SSL is relevant for the further clinical course of the disease, but obese patients do not present the typical peritonitis picture [8]. Due to the extensive visceral fat mass, the infection is initially captured and does not spread diffusely. This aspect significantly delays the detection of SLL in obese patients [8]. The time of first diagnosis of SLL determines the further therapeutic procedure. In early SLL re-laparoscopy, lavage and super sewing of the insufficiency or endoscopic techniques for primary wound closure can be performed [9,10]. In detected SLL two days after primary surgery super sewing is not promising. Secondary wound healing techniques, such as endoscopic stent therapy or endoscopic negative pressure therapy (ENPT), are used in these cases [11].

ENPT is an effective and precious tool in the management of surgical complications after surgery of the gastrointestinal tract [12]. A drainage wrapped by an open-pore suction device (OPSD) is placed endoluminal in position of the leak or intracavitary. Via the drain a negative pressure is applied and causes wound cleansing, defect closure and tissue granulation [13]. OPSD are used as either a polyurethane sponge connected to a drain—a so called open-pore polyurethane foam drainage (OPD), or as a thin open-pore double-layered drainage film (OFD), which is hand-wrapped around a gastric tube14. An advantage of the OFD is its small outer diameter and its possible use on enteral feeding tubes for simultaneous enteral feeding and ENPT [14]. See the used OPSD in Figure 1.

The diagnostic gold standard in patients after LSG suspected for SLL is the immediate performance of sectional imaging [15]. In cases of suspected SLL with a small number of air bubbles in the position of Hiss’ ankle, ENPT can lead to healing as a stand-alone therapy. When big fluid and pus collections outside the gastric lumen are visible in the CT scan, laparoscopic or radiological interventional drainage is necessary [16]. 

Combined surgical and endoscopic treatment, known as rendezvous procedure, is used to reduce the number of examinations under general anaesthesia for these critical ill patients. This rendezvous procedure requires increased staffing on the part of the endoscopy department. The benefits of the rendezvous procedure are currently not proven by studies.

## 2. Materials and Methods

### 2.1. Study Design

The local Institutional Review Board approved this study (IRB number: 464/2021BO2). All patients treated in the time between February 2018 and March 2021 using ENPT for SLL after LSG were considered for inclusion in this study, given the following criteria were fulfilled: confirmed diagnosis of SLL and treatment for the complication at our department. Exclusion criteria were treatment without ENPT and treatment of staple line leak outside our hospital. Informed consent was obtained from all individual participants. The focus of this analysis is postoperative complication management, so it included patients who had been operated on in another hospital.

### 2.2. Rendezvous Procedure

During re-laparoscopy for SLL in LSG patients, an endoscopic team consisting of a doctor and a nurse join in the operating room. During endoscopic diagnostic and placement of an OPSD, surgeons perform laparoscopy. The direct visualization of the perforation by air leakage is possible for the surgeon and the endoscopist. The application of the OPSD is carried out in the same way as described below.

### 2.3. One- or Two-Stage Approach

Endoscopic staff team examine the obese patient with suspected or diagnosed SLL after LSG prior or after the secondary surgery. Depending on the clinical course, patients can also be treated exclusively endoscopically. The application of the OPSD is carried out in the same way as described below.

### 2.4. Application of the OPSD

The first diagnostic and therapeutic endoscopy for SSL were realized in endotracheal intubation anaesthesia in all included cases. In the majority of cases an OFD was handmade, as described elsewhere [14,16], by wrapping a very thin open-pore double-layered drainage film (Suprasorb CNP, Drainage Film; Lohmann & Rauscher International GmbH & Co. KG, Rengsdorf, Germany) on the gastric segment of a nasojejunal feeding tube (Freka Trelumina, Fresenius Kabi Deutschland GmbH, Bad Homburg Germany). Sutures (Mersilene, Polyester, 4 Ph. Eur., Ethicon, Norderstedt, Germany) were used for the fixation of drainage film around the tube. The OFD device was placed endoluminal in the gastric sleeve and was manufactured to cover the leak area with an overlap of the healthy stale line sector by 2 cm at minimum to the proximal and distal direction. The distal segment of the tube was used for enteral feeding. The OFD was guide wired pushed through the lumen. 

In one case, with a perforation size of more than 2 cm, the primary OPSD was an OPD. We used the commercially available product ESO-Sponge System (BBraun Melsungen AG, Melsungen, Germany). This was positioned using the loop technique, in which a loop (Mersilene, Polyester, 4 Ph. Eur; Ethicon, Norderstedt, Germany) was fixed at the distal end of the drainage sponge, gripped with an endoscopic grasper, then placed under endoscopic view. 

Drains of the OPSD were oro-nasal redirected and fixed with plasters. After placement of the OPSD drains were connected to an electric vacuum pump (KCI V.A.C. Freedom; KCI USA Inc., San Antonio, TX, USA) and a continuous vacuum of −125 mmHg was generated. 

### 2.5. Follow-Up Procedures

According to the clinical course and the individual risk of the patients, the follow-up examinations were mostly performed under sedation and only rarely under intubation anaesthesia. A diagnostic endoscopy was performed following the removal of the OPSD. Whenever possible, re-endoscopy was performed after 5–7 days in cases treated with OFD and 3–5 days using OPD. In the case of persisting leak or in the case of uncertainty, an OPSD was reinserted, and treatment was continued. 

### 2.6. Data Analysis

Analysis was performed using SPSS v. 24.0.0.1 (IBM, Armonk, NY, USA). Data were presented as means ± SD. Mann–Whitney U test was performed for comparing means when necessary.

## 3. Results

Twelve patients with SLL following LGS were included in this trial. In the observed time span, in sum, 389 patients with obesity were treated with LGS at our centre. The inhouse SSL-rate was 1.54%. Three patients were operated in other hospitals. Patients with SSL were included for the analysis and were divided into two groups: 

Group A: patients treated for SSL by rendezvous procedures and

Group B: patients treated for SSL solely endoscopically or in sequence surgically and endoscopically.

The patient characteristics are shown in Table 1. In Group A, there were three patients included with complications after LGS operated in other hospitals. No patients from other hospitals were listed in Group B. The gender distribution in both groups differed, with more males in Group A. The other preoperative data were the same in both groups. 

On average, SLL was suspected earlier in Group A than in Group B, although there was considerable variation in both groups.

The treatment characteristics are shown in Table 2. At the time of diagnosis of staple line leaks, patients in Group A were characterized by more severe infection and, in some cases, sepsis. In six patients of Group A and two patients of Group B, peri-gastric abscesses were detected. In the intensive care unit, four patients in Group A and two patients in Group B were treated. Time span of ENPT, number of changes of the negative-pressure devices, length of hospital stays and time span of treatment on ICU differed significantly in both groups with longer therapy time in Group A. 

All patients in Group A underwent re-operation at least once. In Group B, surgery was performed in two patients. The other patients in Group B were successfully treated by ENPT solely. 

In one patient of Group B OPSD dislocated, accidentally. No further therapy-associated complications occurred. No case of postoperative stricture was seen.

SLL-therapy was successful in 11/12 patients. One patient in Group B already had extensive cardiomyopathy prior to bariatric surgery and did not recover under therapeutic measures. This patient died due to septic organ failure.

All patients in Group A underwent re-operation at least once. In Group B surgery was performed in two patients. The other patients in Group B were successfully treated by ENPT solely. 

In one patient of Group B, OPSD dislocated accidentally. No further therapy-associated complications occurred. No case of postoperative stricture was seen.

## 4. Discussion

SLL after LGS is a rare but life-threatening complication [1,2]. Patients with obesity often have pre-existing cardiovascular and pulmonary disease and undergo bariatric surgery in a compromised starting condition [3]. In cases of SLL local inflammation is often not detected early because of the high amount of visceral fat, which result in occult peritonitis without typical pain symptoms [4]. Furthermore, the longer an abdominal focus of infection persists untreated, the higher the risk for systemic inflammatory re-emergence in the sense of sepsis. 

In this analysis, the time of clinical suspected SLL was in Mean on the twelfth postoperative day. Most patients presented with fever, abdominal pain, and high elevated inflammatory markers. In SLL, detection after more than 48 h after the bariatric surgery primary wound closure is not sufficient in most cases.

A CT scan is the gold standard in case of suspected SLL after LGS [5]. Depending on the imaging findings, the extent of inflammation, and the presence of an intra-abdominal abscess, the indication for re-laparoscopy for lavage and drainage is given [2]. 

If intra-abdominal abscesses are found, treatment with stent or clip closure of the perforation or fistula is not sufficient. The abdominal focus must be additionally drained radiologically or surgically. The drainage of secretions through an internal drainage by implantation of a double-pigtail-drainage to endoluminal can lead to a successful healing of the insufficiency in up to 78% according to the study results [6,7]. One of the reasons for the better outcome of patients in Group B is caused by the included patients without intra-abdominal abscesses in this group. Patients were less severely ill at the time point of diagnosis of SLL in Group B compared to Group A.

The ENPT is based on an OPSD (e.g., polyurethane sponge), which is either endoluminally inserted at the stage of the leakage or intracavitary placed into the resulting insufficiency cavity. The open-pore element is fixed to a drainage with perforations, which is connected to a vacuum source. The negative pressure acts through the pores on the surrounding tissue and results in a continuous drainage of secretions, cell-detritus and bacteria, the suction induces tissue proliferation, and a decreased wound size [8,9,10]. 

ENPT is also known under the synonyms E-VAC and EVT. For ENPT as primary endoscopic procedure for leakages after bariatric surgery, possibly in combination with laparoscopy; three studies are currently available with a cumulative success rate of 90.27% in a total of 31 patients. In addition, there are numerous case reports and studies, some of which deal with the combined use of ENPT with stent procedures as first and second line therapy. An alternative closure of leakage after bariatric surgery can be successfully performed with OTSC as first or second line therapy with good results up to closure rates of 86.3% [11,12]. The most frequently performed endoscopic therapy for leakages after bariatric surgery worldwide is the stent therapy [1,13,14]. A challenge is the stent fixation in bariatric patients. Stent dislocation is the most common complication of this type of therapy [15]. Special bariatric stents with a big outer diameter and bulbs have been developed [16]. Because of a high dislocation rate and good results of the ENPT we changed our concept of stent-based treatment of SLL to ENPT in 2016. 

In centres that specialize in the treatment of bariatric patients and have round-the-clock endoscopy, the rendezvous procedure is easy to implement. The concept of the rendezvous procedure is to apply a one-stage combined internal and external drainage of the abscess during one examination with endotracheal intubation to avoid reintubation and septic episodes. Especially, in patients with small leaks, a reliable identification of the leak can be made by the combined laparoscopic and endoscopic procedure. 

One patient died because of septic multiorgan failure. This patient with obesity suffered pre-operatively on relevant cardiomyopathy. It must be assumed that the septic shock was so stressful for him that, despite intensive therapeutic measures, the progressive organ failure could no longer be stopped. This case impressively demonstrates that patients with bariatric surgery often suffer from significant systemic diseases and that postoperative complications quickly lead to fulminant organ failure.

The rendezvous procedure is associated with a high level of personnel effort. We wanted to use a retrospective analysis to investigate whether there is an advantage for patients who have been treated by means of a rendezvous procedure. Obese patients with SLL often require intensive care and continuation of invasive ventilation. Alternatively, to the rendezvous procedure, patients can be managed in two stages, undergoing surgery or endoscopy first and the second procedure in close interval. In summary, our analysis shows that in patients with septic complications after bariatric surgery with indication for re-laparoscopy, a simultaneous endoscopy with application of an OPSD for ENPT can be advantageous. However, a significant benefit for the rendezvous procedure is missing. The series is retrospective with a small number of cases and too time-scattered to draw any significant conclusions. A prospective series is to be preferred, given the low incidence of SLL complications.

## 5. Conclusions

We believe that a one- or two-step procedure with surgical and endoscopically interventions in short intervals can be applied as well. In patients without the need of re-laparoscopy, ENPT is an effective treatment tool as stand-alone interventional therapy. 

## Figures and Tables

**Figure 1 jcm-10-05670-f001:**
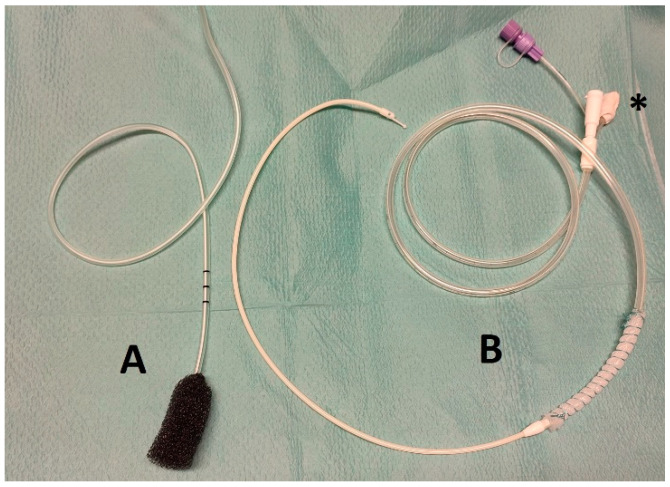
Two OPSD types used in this analysis: **A** = commercially available polyurethane sponge (Eso-Sponge; BBraun Melsungen, Germany); **B** = hand wrapped naso-jejunal feeding tube (Trelumina FREKA, 9 Ch intestinal tube, 16 Ch naso-gastric tube with perforations; Fresenius Kabi Deutschland GmbH, Bad Homburg, Germany) wrapping with cut to size CNP^®^-film (Suprasorb CNP^®^ Drainage Film; Lohmann & Rauscher International GmbH & Co. KG, Rengsdorf, Germany), fixation with suture (Mersilene^®^, Polyester, 4 Ph. Eur; Ethicon—Johnson & Johnson Medical N.V., Belgium). * = closed venting tube.

**Table 1 jcm-10-05670-t001:** Characteristics in patients with (Group A) or without (Group B) rendezvous procedure.

	Group A (*n* = 6)	Group B (*n* = 6)	
Number of male sexes	3	1	n.s.
Mean Age (years)	53.17	52.67	n.s.
Mean BMI (kg/m^2^)	50.37	53.17	n.s.
Mean primary diagnosis of SLL (days after surgery)	8	15	n.s.
Number of detected perigastric abscesses in CT imaging	6	2	n.s.
Mean CRP (mg/dL)	27.09	24.74	n.s.
Mean White Blood Cells (µg/dL)	16,713	12,483	n.s.

Abbreviations: SLL—staple line leaks; CRP—C-reactive protein; n.s.—not significant.

**Table 2 jcm-10-05670-t002:** Treatment characteristics in patients with (Group A) or without (Group B) rendezvous procedure.

	Group A (*n* = 6)	Group B (*n* = 6)	
Mean duration on ICU (days)	6	5	n.s.
Mean number of endoscopic interventions			
-OFD	6	5	n.s.
-OPD	0	1	
Mean number of OPSD changes	5	5	n.s.
Mean duration of ENPT (days)	31	18	n.s.
Mean duration of hospital stay	39.5	20.17	0.047
Number of deceased patients	0	1	

Abbreviations: OFD—open-pore film drainage; OPD—open-pore polyurethane sponge drainage; n.s.—not significant; ENPT—endoscopic negative pressure therapy.

## Data Availability

Data are available on request.

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
