# Peer review of "Role of Rendezvous-Procedure in the Treatment of Complications after Laparoscopic Sleeve Gastrectomy"

_jcm, 2021, doi:10.3390/jcm10235670_

Round 1
Reviewer 1 Report
Well written new method of managing difficult case
Author Response
Thanks for your comments and positive assessment.
Reviewer 2 Report
There are some flaws in this paper:
- The Title may be misleading as this series is referred only to LSG and not "Bariatric Surgery" at large;
- Results and Discussion are somewhat contradictory as it seems that Authors advocate Rendezvous procedure while reported results show that group B pts have better outcomes and fare much better than group A (rendezvous approach)
- The series is retrospective, too limited (12 pts) and too time-scattered to draw any sugnificant conclusion. A prospective series is to be preferred, given the low incidence of SLL complication
- A little remark: Table 1 and 2 refer to groups 1 and 2 while these are named A and B throughout text: this should be amended
Author Response
Thanks for your helpful comments. We changed the named detials (title, labeling in tables and insert a new paragraph into the discussion section). The manuscript has benefited from these changes. Thanks for your effort.
Reviewer 3 Report
Good article on an interesting topic
Author Response
Thanks for your positive assessment.
Round 2
Reviewer 2 Report
There are still some minor flaws to be amended:
- Lines 63-64: ...so called ....(OPD): this has already been defined in line 60
- Lines 181-183 and lines 208-210 report the same sentence. One of the two must be erased.
Author Response
Dear Reviewer, thanks a lot for your comments. We changed the named points.